# Bi$_1$Te$_1$ is a dual topological insulator

Markus Eschbach[1,*], Martin Lanius[1,*], Chengwang Niu[1,2,*], Ewa Młyńczak[1,3], Pika Gospodarič[1], Jens Kellner[4], Peter Schüffelgen[1], Mathias Gehlmann[1], Sven Döring[1,5], Elmar Neumann[1], Martina Luysberg[6], Gregor Mussler[1], Lukasz Plucinski[1,5], Markus Morgenstern[4], Detlev Grützmacher[1], Gustav Bihlmayer[1,2], Stefan Blügel[1,2] & Claus M. Schneider[1,5]

New three-dimensional (3D) topological phases can emerge in superlattices containing constituents of known two-dimensional topologies. Here we demonstrate that stoichiometric Bi$_1$Te$_1$, which is a natural superlattice of alternating two Bi$_2$Te$_3$ quintuple layers and one Bi bilayer, is a dual 3D topological insulator where a weak topological insulator phase and topological crystalline insulator phase appear simultaneously. By density functional theory, we find $\mathbb{Z}_2$ indices (0;001) and a non-zero mirror Chern number. We have synthesized Bi$_1$Te$_1$ by molecular beam epitaxy and found evidence for its topological crystalline and weak topological character by spin- and angle-resolved photoemission spectroscopy. The dual topology opens the possibility to gap the differently protected metallic surface states on different surfaces independently by breaking the respective symmetries, for example, by magnetic field on one surface and by strain on another surface.

[1] Peter Grünberg Institut and JARA-FIT, Forschungszentrum Jülich GmbH, 52425 Jülich, Germany. [2] Institute for Advanced Simulation, Forschungszentrum Jülich GmbH, 52425 Jülich, Germany. [3] Faculty of Physics and Applied Computer Science, AGH University of Science and Technology, al. Mickiewicza 30, Krakow 30-059, Poland. [4] II. Institute of Physics B and JARA-FIT, RWTH Aachen University, Aachen 52074, Germany. [5] Faculty of Physics, University of Duisburg-Essen, D-47057 Duisburg, Germany. [6] Ernst Ruska-Centre for Microscopy and Spectroscopy with Electrons, Forschungszentrum Jülich GmbH, Wilhelm-Johnen-Straße, 52425 Jülich, Germany. * These authors contributed equally to this work. Correspondence and requests for materials should be addressed to L.P. (email: l.plucinski@fz-juelich.de).

Topological insulators (TIs) are bulk insulating materials that exhibit metallic conductivity on their boundaries via electronic edge (in two-dimensional (2D) TIs) or surface states (in three-dimensional (3D) TIs), which are guaranteed by the topology of the bulk band structure[1,2]. Electrons in these boundary states are spin polarized. Their spin and momentum are locked to each other by spin–orbit coupling, creating helical spin textures, which make TIs highly attractive for spintronic applications[3]. One of the most favourable aspects of 3D TIs is the fact that their surface inevitably hosts these metallic surface states as long as the symmetry defining the topological index is not broken[4,5]. In a strong TI (STI), time-reversal symmetry protects these states on all surfaces. Weak TIs (WTIs), on the other hand, display protected metallicity only at surfaces with a certain orientation, while other surfaces do not contain topologically protected surface states. The latter can be understood in a simple picture, where a stack of 2D TIs forms a WTI with metallic surface states inherited from the edge states of the 2D TI but with an insulating surface plane (the dark side) normal to the stacking direction. Finally, in topological crystalline insulators (TCIs), where the symmetry with respect to a mirror plane defines the topology, metallic surface states can be found on surfaces perpendicular to these mirror planes[6,7].

Bi$_2$Te$_3$ was the first material predicted to be both a STI and a TCI. Since it exhibits two topological properties, it was termed a dual TI[8]. Such a combination opens the possibility that controlled symmetry breaking would destroy certain surface states while keeping others intact. For example, one could imagine a material that is both a WTI and a TCI and has all surfaces covered by metallic surface states, that is, the mirror plane of the TCI is normal to the dark side of the WTI. Then, a magnetic field would destroy the topological protection of the states caused by the WTI character, while the mirror-symmetry-protected states remain intact such that the dark surface is metallic. Likewise, small

structural distortions can break the mirror symmetry without affecting the surface states arising from time-reversal symmetry, thus rendering only the dark side insulating.

In the search for such a material, we start from Bi$_2$Te$_3$ with the properties mentioned above and from the Bi bilayer (BL) which is known to be a 2D TI[9–11]. It is possible to produce natural superlattices [Bi$_2$]$_x$[Bi$_2$Te$_3$]$_y$ from hexagonal, metallic BLs and semiconducting Bi$_2$Te$_3$ quintuple layers (QLs) in a wide range of $x$ and $y$[12–15]. Within this series Bi$_2$Te$_3$ consists of QL building blocks only, while the unit cell of Bi$_1$Te$_1$ exhibits a stacking sequence of a single BL interleaved with two subsequent QLs. The size of the unit cell along the stacking direction, that is, the $c$ lattice constant, varies quite severely among the different stable compounds, which makes them easily distinguishable in a diffraction experiment. Recently, a similar superlattice Bi$_4$Se$_3$ (that is, $x = y = 1$) was investigated in some detail and characterized as a topological semimetal[16]. For practical applications, however, an insulating bulk material is preferable.

In this article, we identify the stoichiometric natural super-lattice Bi$_1$Te$_1$ (that is, $x = 1$, $y = 2$) as a semiconductor with a small bandgap of about 0.1 eV with the desired properties: Bi$_1$Te$_1$ is both a WTI and a TCI, and hence a novel type of dual TI with the favourable property of suitable surfaces as described above. Our density functional theory (DFT) calculations predict a $\mathbb{Z}_2$ class of (0;001) and a mirror Chern number $n_{\mathscr{M}} = -2$. We find two characteristic surface states of the TCI on the (0001) surface, regardless of the surface termination. A similar situation has been reported recently for Bi$_2$TeI, theoretically[17]. We demonstrate that Bi$_1$Te$_1$ can be grown in the form of high-quality thin films on Si(111) by molecular beam epitaxy (MBE). Its layered structure is confirmed by scanning transmission electron microscopy (STEM) and X-ray diffraction (XRD), depicting a repeated stacking sequence of 2QL of Bi$_2$Te$_3$ and a single Bi BL. We investigate the electronic structure of Bi$_1$Te$_1$ by means of spin- and angle-

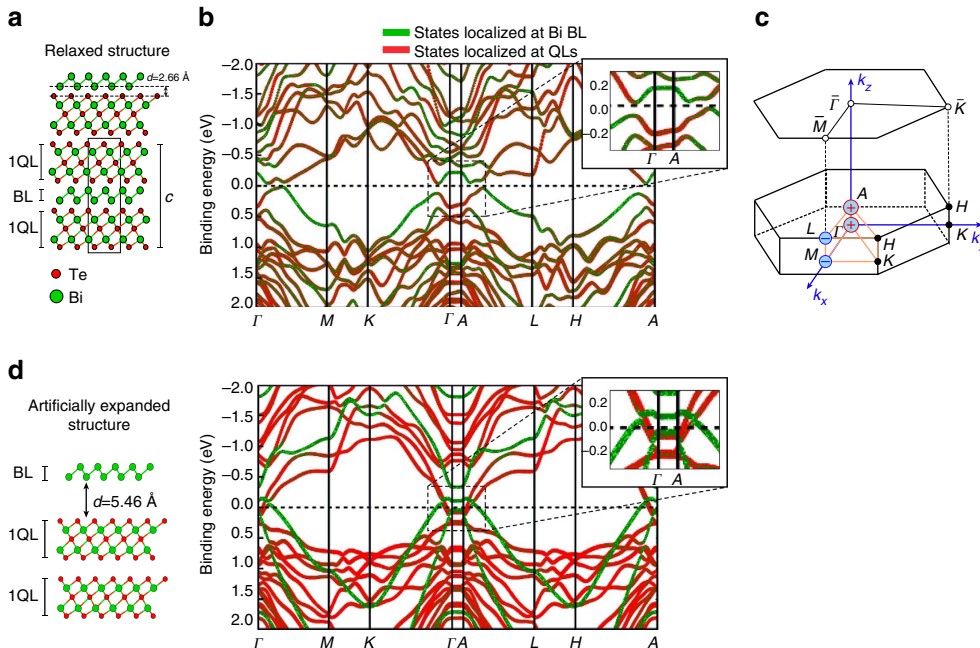

**Figure 1 | Bulk band structures of Bi$_1$Te$_1$. (a)** Simple sketch of the crystal structure of Bi$_1$Te$_1$. The unit cell consists of 1Bi BL and 2QLs. **(b,d)** The bulk band structure calculation in the structurally relaxed geometry **(c)** and with artificially expanded distances between the BL and the QLs **(d)**, respectively. States localized mostly in the BL are marked in green, while the states localized mostly in the 2QLs are shown in red. In **d**, the band structure of the BL shows an inverted gap about 0.2 eV above the Fermi level ($E_F$ marked with a dashed line). **(c)** Bulk and surface Brillouin zone with parity product of the TRIM points for the relaxed structure resulting in +1 (red '+') or −1 (blue '−'). The $k_z$ direction corresponds to the stacking direction. $\Gamma AML$, that is, ($k_x, k_z$)-plane, marks a mirror plane.

resolved photoemission (spin-ARPES) on the (0001) surface. Our spectra taken along non-high-symmetry lines reveal band crossings away from time-reversal invariant momenta (TRIM) points that can be associated with surface states protected by mirror symmetry and the TCI character of $Bi_1Te_1$. Spin-ARPES reveals that the surface states surrounding $\Gamma$ and being close to $E_F$ exhibit a nearly vanishing spin polarization in contrast to the directly compared time-reversal symmetry-driven TSS in $Bi_2Te_3$ ($x = 0$, $y = 3$, $y = 3$ here because the unit cell consists of three QLs), which is in line with WTI character with the dark surface being (0001).

## Results

*Ab initio* calculations. Figure 1a depicts a schematic model of the crystal structure of $Bi_1Te_1$, indicating Bi (green) and Te (orange) atoms as well as one unit cell defined by the lattice constant $c$ along the stacking direction. The separation of the layered structure into QLs and Bi BLs is marked. The bulk band structure of $Bi_1Te_1$ in the relaxed structural geometry is presented in Fig. 1b. Spin–orbit coupling is included in this calculation and the colour represents the localization of the electronic states at the BL (green) or at the QL (red). As one can see, there are no states at the Fermi level, $E_F$, reflecting the insulating character with an energy gap of 73 meV. The states around the Fermi level alternate between BL- and QL-related, where the highest occupied levels at the time-reversal invariant momenta $\Gamma$ and $A$ stem from QLs (red), while the lowest unoccupied states originate from the BL (green), and *vice versa* for the $M$ and $L$ points. Since the crystal possesses spatial inversion symmetry, the parity of the states can be calculated and the topological index $\mathbb{Z}_2$ can be deduced according to ref. 18 based on the product of the parities of all occupied bands at the eight TRIMs, that is, one at $\Gamma$, one at $A$, three at $M$ and three at $L$. The result is shown in Fig. 1c for the corresponding TRIMs, $\Gamma$ and $A$ have parity products of $+1$ (red '+') while $M$ and $L$ have $-1$ (blue '−'), leading to a topological invariant $\mathbb{Z}_2 = (0;001)$. Therefore, $Bi_1Te_1$ is a weak TI with the (0001) surface, which is perpendicular to the stacking direction, being the dark surface and being free of time-reversal symmetry-protected surface states.

It is tempting to relate the WTI property to the fact that both the BL and the 2QLs $Bi_2Te_3$ are 2D TIs such that the WTI results from a simple stacking of 2D TIs in the $c$-direction. However, our band structure calculations in Fig. 1d, which introduce artificially expanded distances between the BL and the QLs, show a more complex scenario: if the BL is sufficiently separated from the 2QLs, the states can be decomposed into contributions from the two components (green = Bi BL and red = 2QLs, respectively). However, due to charge transfer, the inverted gap of the BL is shifted above the Fermi level and, accordingly, some of the 2QL $Bi_2Te_3$ conduction band states are below $E_F$. Only the hybridization of the BL states with the QL states opens up the gap that leads to the insulating bulk structure in Fig. 1b, as can be deduced from the changing colour of the bands along the $k$-directions. Nevertheless, the topological character of the stacked film remains non-trivial. A similar complexity is also found for the first confirmed, stacked weak TI $Bi_{14}Rh_3I_9$ (refs 19–21).

Next, we examined the surface band structure of the (0001) surface, which heavily depends on the precise surface termination. Due to the layered crystal structure and the weak van der Waals bonds between the subsequent building blocks, there exist three natural cleavage planes and thus surface terminations, that is, 1Bi BL, 1QL and 2QL. Figure 2 depicts the spin-resolved surface band structure for the latter two terminations. In all cases an even number of Fermi level crossings is found in $\overline{\Gamma M}$ direction (see Supplementary Notes II and III for all terminations and

convergence with film thickness). We observed that, although there are surface states, the band structure is compatible with the fact that we look at the dark surface of a WTI phase. Nevertheless, it is remarkable that the bands along $\overline{\Gamma M}$ show a band crossing for all possible terminations, reminiscent of Dirac-like cones observed in TCIs. Additional evidence that this crossing is protected by a mirror symmetry in the crystal comes from the observation that the crossing is lifted when the surface atoms are displaced in $[1\bar{1}00]$ direction, breaking this symmetry in one of the $\overline{\Gamma M}$ directions (Fig. 2c).

To finally confirm that $Bi_1Te_1$ is a TCI, we determined the mirror Chern number of the bulk phase. In the $(k_x, k_z)$ plane in reciprocal space (Fig. 1c) all Bloch states can be distinguished by their eigenvalues with respect to a mirror operation in the $(1\bar{1}00)$ plane. To calculate their corresponding Berry phases as well as the Chern numbers, we construct a tight-binding Hamiltonian based on the maximally localized Wannier functions[22]. The Chern numbers of all occupied bands for the opposite mirror eigenvalues $+i$ and $-i$ are $n_{+i} = 2$ and $n_{-i} = -2$, respectively, and therefore the mirror Chern number $n_{\mathscr{M}}$ (ref. 6), given as $n_{\mathscr{M}} = (n_{-i} - n_{+i})/2$, is $n_{\mathscr{M}} = -2$, confirming the fact that $Bi_1Te_1$ is a TCI. The modulus of $n_{\mathscr{M}}$ shows that we have to expect two linear crossings along the high-symmetry line formed by the mirror- and the surface-plane[23], the sign determines the spin orientation[6] shown in the Supplementary Note II. The mirror Chern number $n_{\mathscr{M}} = -2$ is incompatible with STI phase since it produces even number of crossings. In this context it might be interesting to note that the dual TI $Bi_2Te_3$ has $n_{\mathscr{M}} = -1$, and consequently a single Fermi level crossing[8].

Let us discuss the individual features of the different terminations shown in Fig. 2 in more detail. The QL-terminated surface (Fig. 2b) shows near the Fermi level only the linear crossing of bands, which is protected by mirror symmetry. In the 2QL case, we observe, in addition, an upward-dispersing, Rashba-type spin-split parabolic surface band with 0.15 eV binding energy at $\overline{\Gamma}$. It touches the bulk valence band along $\overline{\Gamma K}$ and hybridizes with the protected surface state around $-0.1$ eV in $\overline{\Gamma M}$ direction. The BL-terminated surface shows characteristic downwards dispersing states, very similar to features observed for a single BL on $Bi_2Te_3$ (refs 24,25), that are not seen in the experimental spectra shown below (calculations are presented in the Supplementary Notes II and III). All terminations show strongly spin-polarized surface states around 1.0 eV (Fig. 3a), which are similar to the Rashba-type surface states that also characterize the surface of $Bi_2Te_3$ (ref. 26) or $Sb_2Te_3$ (ref. 27).

**Crystallographic structure.** Figure 4 shows the experimental characterization of the bulk crystal structure of our $Bi_1Te_1$ thin films via XRD (Fig. 4a–c) and STEM (Fig. 4d). From the $\omega/2\theta$ scans in Fig. 4a, the crystal phase was determined by comparing the peak positions with the calculated Bragg reflections, both for $Bi_1Te_1$ and $Bi_2Te_3$. The in-plane and out-of-plane lattice constants $a$ and $c$ were determined precisely from the reciprocal space maps around the $(1,0,-1,20)$ reflection for $Bi_2Te_3$ (Fig. 4b) and the $(1,0,-1,16)$ reflection for $Bi_1Te_1$ (Fig. 4c). The peaks were fitted with Gaussians for determination of the experimental error. We find $a = 4.37 \pm 0.12$ Å and $c = 30.51 \pm 0.46$ Å for $Bi_2Te_3$ and $a = 4.45 \pm 0.02$ Å and $c = 24.0 \pm 0.1$ Å for $Bi_1Te_1$.

In addition, the stoichiometries of the samples were also checked by Rutherford backscattering spectroscopy which confirmed the 50:50 ratio of Bi:Te (see Supplementary Fig. 1).

Figure 4d depicts a high-angular annular dark field image of a representative section of a 39-nm-thick $Bi_1Te_1$ film recorded by STEM. The observed clear contrast is related to the difference between individual atomic columns of Bi and Te (Bi atomic

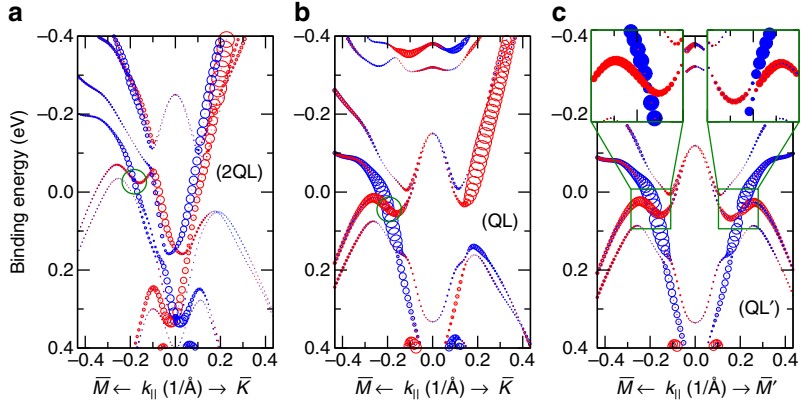

**Figure 2 | Spin-resolved DFT surface electronic structure calculations.** Band structure along $\overline{M\Gamma K}$ of slabs of $Bi_1Te_1$ terminated by (**a**) 2QLs and (**b,c**) 1QL. The size of the symbols corresponds to the spin polarization in the first four layers of a slab, the colour (red/blue) indicates the orientation of the spins with respect to a direction perpendicular to the momentum and surface normal. The green circle marks the Dirac cone of the TCI. In **c** (QL'), the symmetry has been broken by strain along [$1\bar{1}00$] direction, such that $\overline{\Gamma M'}$ is no longer a high-symmetry line and a gap opening can be observed (see inset).

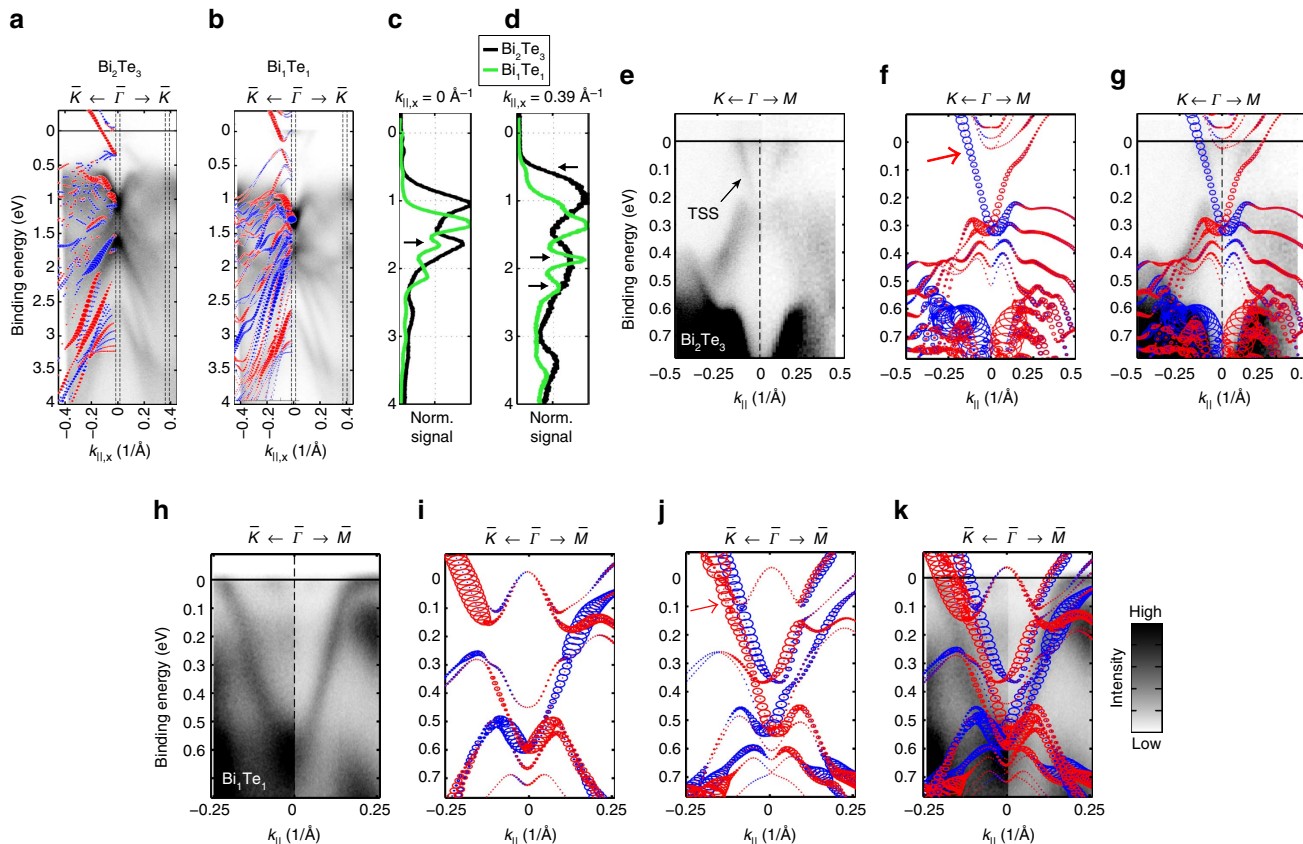

**Figure 3 | Band structure of $Bi_2Te_3$ and $Bi_1Te_1$.** Comparative wide energy range ARPES spectra along $\overline{\Gamma K}$ direction of thin films of (**a**) $Bi_2Te_3$ and (**b**) $Bi_1Te_1$ measured at $T \approx 25$ K using $hv = 21.2$ eV. The colour code in the experimental spectra scales from bright = low to dark = high intensity. Superimposed are the results from the corresponding spin-polarized DFT calculations (1QL termination from Fig. 2 is used; red and blue dots mark opposite in-plane spin channels). (**c,d**) Energy distribution curves obtained along the dashed areas in **a,b**, respectively, at (**c**) $k_{\|,x} = 0$ Å$^{-1}$ and (**d**) 0.39 Å$^{-1}$ (black curve from $Bi_2Te_3$ and green curve from $Bi_1Te_1$). Black arrows mark prominent spectral differences. Magnified electronic structure close to the Fermi level along indicated crystallographic directions are shown in **e** for $Bi_2Te_3$ at $hv = 21.2$ eV and in **h** for $Bi_1Te_1$ at $hv = 8.4$ eV. Additionally, the corresponding calculations are shown in **f** for $Bi_2Te_3$, and in **i–j** for $Bi_1Te_1$ ((**i**) 1QL termination, (**j**) 2QL termination), and superimposed on the experimental spectra in **g,k**, respectively ((**k**) shows overlap with (**i**) and (**j**)). The Fermi energy in the calculated spectra was shifted up by 100 meV (200 meV) in the 1QL (2QL) $Bi_1Te_1$ case compared to Fig. 2, and by 250 meV for the $Bi_2Te_3$ case. Red arrows point to a single spin-polarized band in $Bi_2Te_3$ (**f**), and two overlapping bands with opposite spin polarization for $Bi_1Te_1$ (**j**).

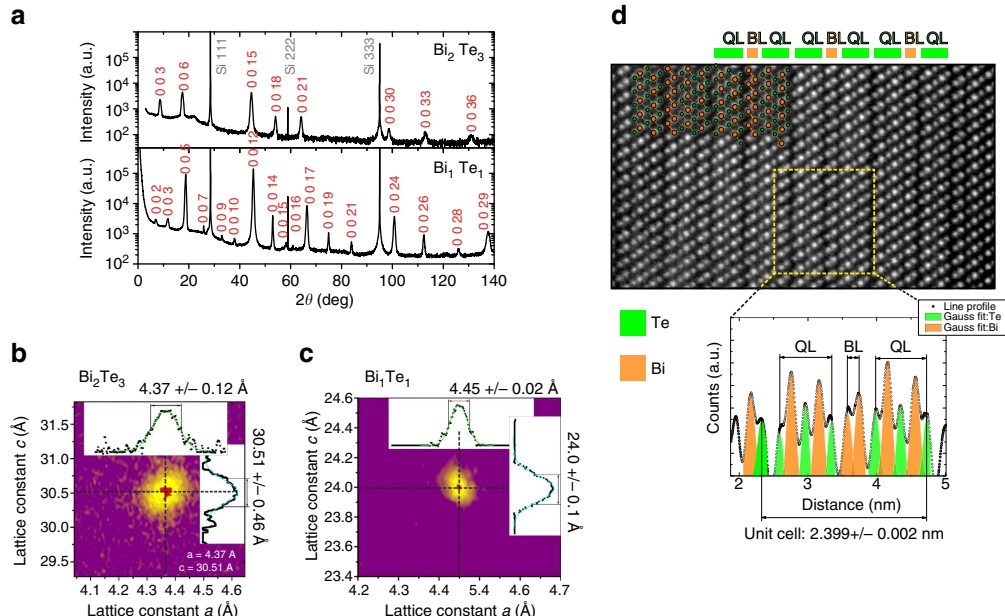

**Figure 4 | Bulk characterization of $Bi_1Te_1$ thin films.** (**a**) Comparative XRD $\omega/2\theta$ scans for both $Bi_2Te_3$ and $Bi_1Te_1$ films averaged over the entire crystal (**b,c**). 2D reciprocal space maps obtained around the $(1,0,-1,20)$ reflection for $Bi_2Te_3$ (**b**) and the $(1,0,-1,16)$ reflection for $Bi_1Te_1$ (**c**) with derived lattice constants $a$ and $c$ from Gaussian fits. (**d**) Local STEM image of a 39-nm-thick $Bi_1Te_1$ film confirming the high bulk crystalline quality. The contrast in the image scales with the atomic number squared ($Z^2$), that is, bright = Bi, darker = Te. QLs and Bi BLs separated by van der Waals gaps can be identified. The yellow frame marks the region over which the line profile below is measured while averaging in the vertical direction. QLs and BLs are denoted and Bi and Te atomic layers are displayed by orange and green columns, respectively.

columns appear brighter than Te columns). Distinct van der Waals gaps, separating QLs from BLs, are visible and the arrangement of BL and QL matches the expected 1:2 composition ratio. Furthermore, by extracting a line profile (yellow frame, which also defines the scale of the image) and fitting Gaussians to the peaks (green = Te; red = Bi) the atomic positions can be determined precisely. Using this method, the size of one bulk unit cell was confirmed to be $c = 23.99 \pm 0.02$ Å, which is in good agreement with the results from XRD.

As we have seen, due to the superlattice character of $Bi_1Te_1$, there is more than one possible surface termination, but neither XRD nor STEM probe the surface. Supplementary Notes II present a spectroscopic study of the chemical composition of the surface of $Bi_1Te_1$ and the influence of noble gas sputtering on the surface termination. It turns out that our growth conditions result in Bi-poor surfaces (that is, a higher amount of QL-terminated surface regions), while ion sputtering leads to Bi-rich surfaces (that is, a higher amount of BL-terminated surface areas). This is the reason why, in the following, we will mostly focus on 1QL- and 2QL-terminated $Bi_1Te_1$, since the samples that have been investigated by high-resolution ARPES (HR ARPES) were vacuum-transferred after growth and have never been exposed to ambient conditions.

However, since we expect differently terminated surface terraces to be in the order of few micrometer in size[16,28], and we employ beam spot sizes of 400 μm (HR ARPES) or even 1 mm (spin-ARPES) in the ARPES experiments, we have to assume that our electronic structure investigations always probe a superposition of different terminations. Therefore, due to the rich variety of surface-related states (Fig. 2) and the fact that our ARPES technique averages over the beam spot size, a detailed distinction of the surface electronic features is challenging.

**Surface electronic structure by ARPES.** The comparative results of our ARPES investigations on vacuum-transferred, as-grown

$Bi_2Te_3$ and $Bi_1Te_1$ thin films are summarized in Fig. 3. In the case of the prototypical STI $Bi_2Te_3$ our results reproduce earlier findings[26]. In general, the spectra exhibit sharp features and a good signal-to-noise ratio revealing the high crystalline quality of the thin films. Figure 3a,b depict wide range binding energy $E_B$ versus wavevector $k_{\|,x}$ maps of $Bi_2Te_3$ (a) and $Bi_1Te_1$ (b), respectively, along trajectories in the $\overline{\Gamma K}$ direction which traverse the $\overline{\Gamma}$ point of the surface BZ, recorded with $hv = 21.2$ eV. Both samples are of $n$-type nature with the conduction band minimum being cut by the Fermi level.

On the first glance the spectra of the two samples show a lot of similarities but a closer analysis reveals some differences, as it can be best seen in energy distribution curves (Fig. 3c,d), which were plotted for $k_{\|,x} = 0$ Å$^{-1}$ (Fig. 3c) (normal emission) and for $k_{\|,x} = 0.39$ Å$^{-1}$ (Fig. 3d). The $Bi_1Te_1$ spectra globally seem to be downshifted which we attribute to a possible electron donation of the BLs to the QLs[16].

The spin-polarized surface electronic structure slab-calculations are superimposed onto the ARPES maps (for 1QL surface termination in the case of $Bi_1Te_1$). The Fermi level in the calculation needed to be shifted upwards by 250 meV (100 meV) to fit better to the experimental data of $Bi_2Te_3$ ($Bi_1Te_1$). The prominent and intense Rashba-type surface state located between $E_B = 0.7$–1.05 eV (0.95–1.3 eV) has been used as a gauge to match the calculation to the ARPES data. As one can see, the agreement between the data and DFT simulation is reasonable and most of the features can be matched.

Figure 3e–g, h–k show the comparison of the magnified electronic structure close to the Fermi level for $Bi_2Te_3$ and $Bi_1Te_1$, respectively. In $Bi_1Te_1$ (Fig. 3h) the predicted gap opening along $\overline{\Gamma K}$ direction for 1QL termination is not reproduced in the experiment. However, the superposition (Fig. 3k) of the calculated spectra of 1QL (Fig. 3i) and 2QL-terminated $Bi_1Te_1$ (Fig. 3j) agrees well with the experimental spectrum where 2QL closes the gap. The gap opening in the surface state along $\overline{\Gamma K}$ is expected since there is no mirror symmetry protecting the

bands along this direction. Note, that the Fermi level in the 2QL-terminated calculation was shifted by roughly 100 meV more than in the 1QL case pointing to different charge transfer. Supplementary Note II depicts wider energy range simulations where one can see that the Fermi level of the 2QL-terminated case indeed needs to be shifted further to match the deeper lying intense Rashba-type features. Additionally, in Supplementary Note IV we analyse the photon energy dependence of the states closest to $E_F$ in both $Bi_2Te_3$ and $Bi_1Te_1$. We find in both cases a negligible out-of-plane dispersion that reveals the surface state character of these features.

In summary, we believe that we do not find a gap-opening in the ARPES data of the surface band close to $E_F$, due to the lack of lateral resolution of our measurement technique, probing different terminations simultaneously.

A strong experimental evidence of the topological nature of a state is the verification of the helical spin polarization of a single surface state[2]. This becomes apparent for the bands marked by red arrows close to $E_F$ (Fig. 3f,j), where two bands of opposing

chirality nearly overlap in the case of $Bi_1Te_1$. Figure 5 summarizes our findings on the spin polarization of the interesting states close to $E_F$ of sputtered, that is, Bi-rich, $Bi_1Te_1$ (Fig. 5a–c). Again the data from $Bi_1Te_1$ is compared to measurements on $Bi_2Te_3$ and the spin polarization of the prototypical TSS (Fig. 5d–f).

Figure 5a,d show the wide range ARPES maps of $Bi_1Te_1$ and $Bi_2Te_3$ from Fig. 3a,b, respectively, which illustrate along which opposing $k$-points, marked by the red dashed area, the spin polarization is measured. Figure 5b,c as well as Fig. 5e,f depict the wide range and near-Fermi level (in-plane) spin-resolved partial intensities $I_{left}$ and $I_{right}$ along the indicated $k$-points. The spectra were corrected by the asymmetry function, sometimes called the Sherman function, of $S = 0.27$, and the net spin polarization is shown underneath. Both samples show quite similar and rather high in-plane spin polarization of 40–50% in the bands at higher binding energies, around $E_B \approx 3.2$, $\approx 2.1$ and $\approx 0.9$–1.1 eV in Fig. 5b,e. The full reversal of the spin polarization between the two opposing $k$-points confirms the helical nature of these states in both samples. Further, the TSS of $Bi_2Te_3$ shows a helical spin

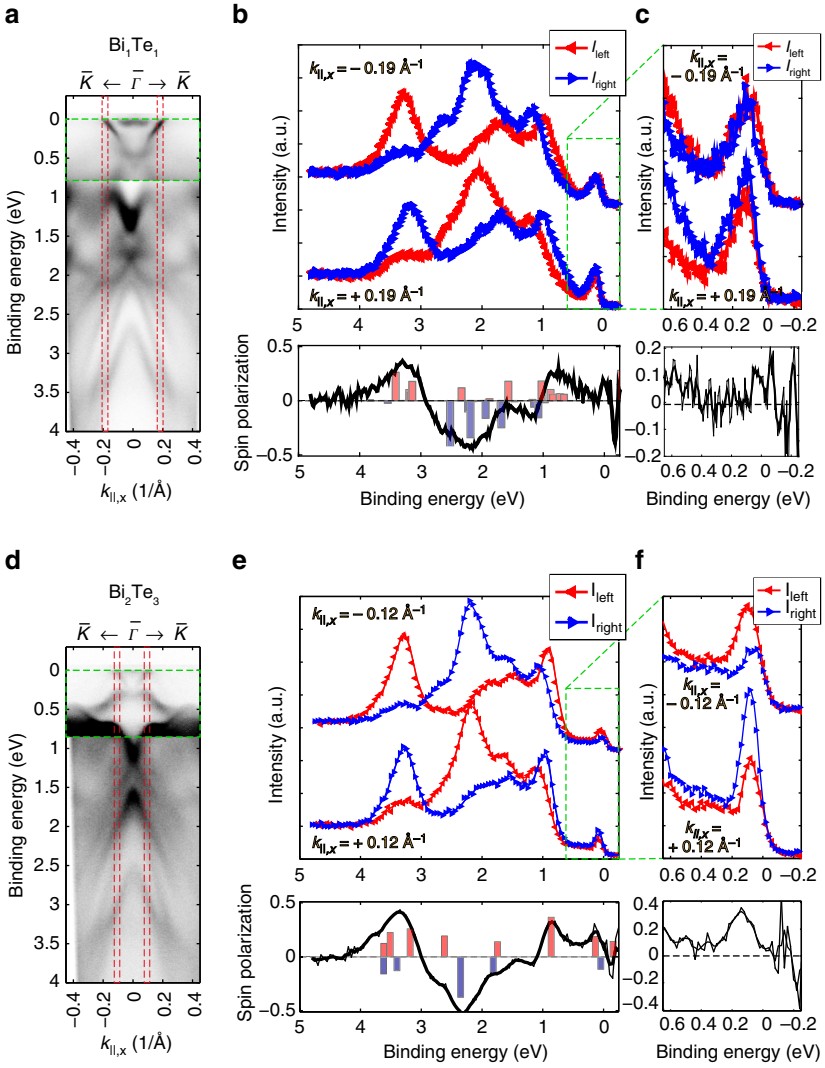

**Figure 5 | Spin-ARPES measurements recorded with $h\nu = 22$ eV and a four-channel SPLEED polarimeter.** Wide range ARPES spectra of (**a**) $Bi_1Te_1$ and (**d**) $Bi_2Te_3$ as shown in Fig. 3b,a for the illustration of the $k$-points $k_{||,x} = \pm 0.19$ Å$^{-1}$ and $k_{||,x} = \pm 0.12$ Å$^{-1}$ (red dashed lines) along which the in-plane spin polarization was measured. Note that the first 800 meV below the Fermi level were boosted in contrast (green dashed area). (**b,e**) Wide energy spin-polarized EDCs at the indicated two opposing $k$-points, and effective net spin polarization below. The bar graph in the latter shows the calculated spin polarization. **c,f** Magnified EDCs and spin polarization close to the Fermi level according to area marked in green in **b,e**.

polarization of up to 40% in Fig. 5f, which confirms its topological nature and is in agreement with what was reported earlier[26]. On the contrary, Fig. 5c reveals that the most interesting states in $Bi_1Te_1$ at the Fermi level exhibit only very small (although non-vanishing) in-plane spin polarization of max. 10% without a clear reversal at the opposing $k$-points. Such weak spin polarization is expected due to spin–orbit coupling (Rashba effect) in the topologically trivial surface states (see states in the calculation in Fig. 3b,c). Therefore, this measurement reveals clear difference to the prototypical TSS and thus gives a strong experimental indication but no final proof for the non-topological character of these states at $E_F$ in $Bi_1Te_1$.

Finally, we describe our experimental evidence for the mirror-symmetry-protected band crossings at non-TRIM points, which are a consequence of the topological crystalline character of $Bi_1Te_1$. Figure 6 depicts experimental and calculated spectra along non-high-symmetry lines which reveal a region in $k$-space, where the TCI-induced states can be identified and no other states interfere. Figure 6a shows a constant energy contour, that is, $k_{||,x}$ versus $k_{||,y}$ map, at $E_B = E_F$ with red dashed lines marking ARPES spectra taken along different cut directions at $k_{||,y} = 0.087\,\text{Å}^{-1}$ (Fig. 6b), $0.120\,\text{Å}^{-1}$ (Fig. 6c), $0.152\,\text{Å}^{-1}$ (Fig. 6d), $0.184\,\text{Å}^{-1}$ (Fig. 6e), $0.216\,\text{Å}^{-1}$ (Fig. 6f) and $0.258\,\text{Å}^{-1}$ (Fig. 6g). Figures 6h–m depict the corresponding spin-polarized surface electronic structure calculations, which are performed for a 1QL-terminated sample. Besides some additional features in the experimental spectra at 0.4–0.6 eV, the agreement is good and one can identify most bands close to the Fermi level. Importantly, we find the mirror-symmetry-protected crossing point around $k_{||,y} = 0.184\,\text{Å}^{-1}$ and $E \approx 150\,\text{meV}$, which exhibits a gap opening after minor symmetry breaking (see Fig. 2c). Additionally, Fig. 6n depicts the energy distribution curve along the red dashed line in spectrum (Fig. 6g), which shows the gap opening away from the Dirac point in agreement with the DFT results. We attribute these crossing bands at a non-TRIM point to be the consequence of the TCI character of $Bi_1Te_1$.

For the $k_{||}$ range presented in Fig. 6, the calculation for 1QL-terminated $Bi_1Te_1$ predicts a region without any states at higher

binding energies between 400 and 600 meV. In the spectra at Xe 8.4 eV excitation, the experimental spectral weight in that binding energy range is indeed decreased; however, one can still identify dispersive features crossing this region. This is related to a small portion of the surface exhibiting termination other than 1QL or 2QL, which we describe in Supplementary Note V and the related discussion there. There, we show that for the probed $k_{||}$ range the spectral weight of the states between 400 and 600 meV strongly depends on the photon energy-dependent photoemission matrix element and nearly vanishes at He I 21.2 eV excitation. In the related theoretical calculations, this region without bands is still present for the 2QL termination, but not for the Bi BL termination.

## Discussion

In summary, we predicted by DFT and demonstrated by ARPES the dual TI character of the stoichiometric natural superlattice $Bi_1Te_1$.

Our study theoretically predicts by *ab initio* DFT calculations that $Bi_1Te_1$ exhibits a dark surface perpendicular to the stacking direction which is free of time-reversal symmetry-protected surface states at the TRIM points, due to weak topological $\mathbb{Z}_2$ indices (0;001). Moreover, we identify an additional protection of topological states with crossings at non-TRIM points in the $\overline{\Gamma M}$ mirror plane direction governed by the crystal mirror symmetry with non-zero mirror Chern number $n_{\mathscr{M}} = -2$. This dual WTI and TCI character leads to the existence of topological states on every surface of the crystal, which are protected either by time reversal or by mirror symmetries, respectively.

Confronting the theoretical predictions with the experiment, we successfully realized thin films of $Bi_1Te_1$ on Si(111) by MBE growth, characterized the bulk crystal structure as well as the surface chemistry, and thoroughly investigated the (spin-) electronic structure. We identified significant differences to the prototypical STI $Bi_2Te_3$, that is, the spin polarization of the surface-related features at the Fermi level is negligible for $Bi_1Te_1$, which points to a non-topological character of this state, and,

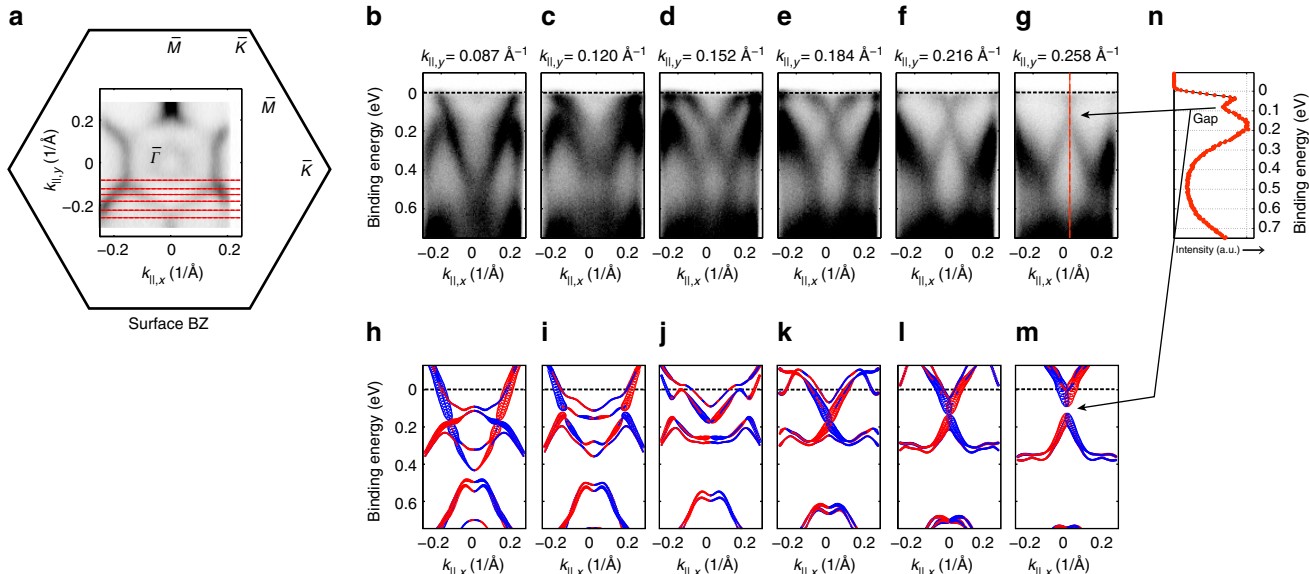

**Figure 6 | Electronic structure of $Bi_1Te_1$ along non-high-symmetry lines revealing TCI surface states.** (**a**) Fermi surface $k_{||,x}$ versus $k_{||,y}$ map of a 45 nm *in situ* transferred film of $Bi_1Te_1$ obtained with $hv = 8.4$ eV. Red dashed lines mark trajectories along which $E_B(k_{||,x})$ spectra in **b–g** were recorded. Sketch of the surface Brillouin zone is marked for orientation. (**b–g**) Corresponding experimental spectra as well as spin-polarized DFT calculations (**h–m**) of a 24QL slab of $Bi_1Te_1$ with 1QL-terminated surface at (**b,h**) $k_{||,y} = 0.087\,\text{Å}^{-1}$, (**c,i**) $0.120\,\text{Å}^{-1}$, (**d,j**) $0.152\,\text{Å}^{-1}$, (**e,k**) $0.184\,\text{Å}^{-1}$, (**f,l**) $0.216\,\text{Å}^{-1}$ and (**g,m**) $0.258\,\text{Å}^{-1}$, respectively. (**n**) Energy distribution curve along the red dashed line in **g**.

hence, is an indication of the WTI nature of $Bi_1Te_1$. Furthermore, we reveal mirror-symmetry-protected band crossings at non-TRIM points in excellent agreement to DFT calculations, which we attribute to the TCI character of $Bi_1Te_1$.

In future work, the weak topological nature of $Bi_1Te_1$ could be confirmed by probing the topologically protected one-dimensional electron edge channels at step edges of the dark surface, for example, by STS study (similar to what was reported in refs 19,29,30).

The dual topological character opens up new vistas because topological states are protected by different symmetries and can be potentially switched on and off separately by breaking one of the symmetries.

## Methods

**Sample growth.** All samples for this study were grown as thin films via MBE. First, $10 \times 10$ mm$^2$ Si(111) samples were prepared by a standard set of wafer cleaning steps (RCA-HF) to remove organic contaminations and the native oxide. A consecutive HF dip passivates the Si surfaces with hydrogen for the transfer into the MBE chamber (base pressure $(5 \times 10^{-10}$ mbar). To desorb the hydrogen from the surface, the substrates were heated up to 700 °C for 10 min and finally cooled down to 275 °C. For the evaporation of Te and Bi, standard effusion cells were heated to $T_{Te} = 260$ °C and $T_{Bi} = 460$ °C, resulting in a growth velocity of $Bi_1Te_1$ of $v = 2.5$ nm h$^{-1}$. The tellurium shutter was opened several seconds in advance to terminate the silicon surface by Te, which saturates the dangling bonds. While $Bi_2Te_3$ is grown in a tellurium overpressure regime[31], $Bi_1Te_1$ requires equal vapour pressures of tellurium and bismuth. The 1:1 ratio between bismuth and tellurium changes the structure from solely QLs in $Bi_2Te_3$ into a layered structure with additional BLs between every two QLs in $Bi_1Te_1$.

After growth, the samples were transferred from the MBE chamber into the ARPES apparatus ($< 1 \times 10^{-10}$ mbar) without breaking the vacuum, by a ultra-high vacuum shuttle with a base pressure below $1 \times 10^{-9}$ mbar. The surface of such as-grown samples is, due to the growth mode, expected to be Bi-poor, that is, mostly QL-terminated. Nevertheless, the surface exhibits all three different terminations (see Supplementary Note II).

**Structural characterization.** For characterizing the bulk crystal structure, XRD measurements were carried out, employing a high-resolution Bruker D8 diffractometer. Additionally, cross-sectional specimens were measured in an aberration-corrected STEM with an electron beam of 0.8 Å (FEI Titan 80–200) for structural investigations on the atomic scale. For this, selected specimens are prepared by focused ion beam etching with firstly 30 keV and subsequently 5 keV Ga ions. Later Ar ion milling using the Fishione NanoMill was performed to reduce the FIB-induced damage. High-resolution STEM images made in high-angular annular dark field contain chemical information, since the contrast scales approximate with the square of the atomic number $Z^2$, allowing to distinguish between Bi and Te atoms.

**Spectroscopy.** The lab-based high-resolution ARPES investigation was performed at $T = 25$ K with an MBS A1 electron spectrometer, using either a non-monochromatized He I$\alpha$ radiation of $h\nu = 21.2$ eV from a focused helium lamp (Focus HIS 13) or light from a microwave-driven xenon discharge lamp (MBS) producing $h\nu = 8.4$ eV photons. The beam spot size is about 400 µm in the former and 1 mm in the latter case and the light is unpolarized. The analyser measures $E_B$ versus $k_{\parallel,x}$ dispersion maps at once. Fermi surface mapping is achieved by rotating the sample with respect to the entrance slit of the spectrometer. The overall energy resolution is estimated to be 10 meV and the angular resolution is $< 0.02$ Å$^{-1}$.

For spin-resolved ARPES measurements we used photons of $h\nu = 22$ eV, a Scienta SES-2002 spectrometer, and a Focus SPLEED polarimeter at beamline BL5 of the DELTA synchrotron in Dortmund with the sample kept at room temperature, resulting in an energy resolution of $\approx 100$ meV[32]. Here, clean sample surfaces were prepared by sputtering and annealing after sample transfer through air, which resulted in Bi-rich sample surfaces (see Supplementary Note II).

**Electronic structure calculations.** The DFT calculations are performed for the bulk phase and thin films with three different surface terminations, namely a single and a double QL (1QL and 2QL), and a Bi BL. The bulk unit cell consists of two QLs and one Bi BL, and the hexagonal atomic planes are all assumed to have a face-centred cubic-like (A–B–C) stacking. To simulate a Bi BL-terminated surface, a symmetric 26 layer film with BL–QL–QL–BL–QL–QL–BL stacking was used. For the 1QL and 2QL termination, symmetric 24 and 34 layer films were set up. We further checked that slabs of 72 and 84 layers did not change the topological character. We employ the full-potential linearized augmented plane wave method as implemented in the Fleur code (see http://flapw.de) with the relaxed lattice parameters from the Vienna *ab initio* simulation package[33,34]. The generalized gradient approximation of Perdew–Burke–Ernzerhof form[35] is used for the

exchange correlation potential. Spin–orbit coupling is included self-consistently in the calculations.

From the DFT calculations, we obtain structural parameters that are in good agreement with the experimental data. The size of the bulk unit cell in *c*-direction is 25.0 Å. It consists of two QLs of 7.48 Å thickness each and a Bi BL of 1.68 Å. The BL-QL separation is 2.66 Å and the distance between the QLs is 3.04 Å. At the surfaces, these distances contract slightly, for example, the QL–QL distance decreases by 0.06 Å at the 2QL-terminated surface, while the QL–BL distance is reduced only by 0.04 Å for the 1QL termination. For the BL termination, the interlayer distance changes even less. The step height between a BL-terminated and a 2QL-terminated surface is thus $1.68 + 2.66 = 4.34$ Å.

It is well known that generalized gradient approximation overestimates the bond length in van der Waals-bonded systems. Therefore, we relaxed the structure also using the DFT-D2 method[36] and obtained a slight contraction of the unit cell in *c*-direction. The resulting lattice constant of 24.0 Å is in good agreement with the experimental value while the topolocical features are unchanged.

**Data availability.** The data that support the findings of this study are available from the corresponding author on request.

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

## Acknowledgements

The authors acknowledge financial support from the priority program SPP1666 of the DFG (projects MU3187/3-1, BI823/2, and MO858/13-2) and the Virtual Institute for Topological Insulators (VITI) of the HGF. We thank Volkmar Hess, Samuel Königs-shofen, Frank Matthes and Daniel Bürgler for additional characterization of the surface chemistry of our samples in their laboratory-based XPS chamber. We also thank B. Holländer for the precise determination of the samples stoichiometry via RBS. Further, the authors acknowledge the technical support by B. Küpper and A. Bremen. The authors gratefully acknowledge the computing time granted on the supercomputer JURECA at Jülich Supercomputing Centre (JSC).

## Author contributions

M.E., Ma.La. and C.N. contributed equally to this work. Ma.La., P.S. and G.M. grew the samples via MBE. M.E., E.M., P.G., J.K. and L.P. carried out the lab-based HR ARPES experiments. C.N., G.B. and S.B. provided the ab initio DFT band structure calculations. M.E., M.G. and S.D. performed spin-resolved ARPES measurements at BL5 of DELTA in Dortmund. Ma.La., E.N. and Ma.Lu. prepared the FIB lamellas and performed HR STEM measurements. G.M. characterized the samples by XRD. B.H. measured RBS for stoichiometry analysis. M.E., L.P., C.N. and G.B. wrote the paper with contributions from all coauthors. The project was supervised by M.M., S.B., D.G. and C.M.S.

## Additional information

**Competing interests:** The authors declare no competing financial interests.

