## [Peer Review File · Nature Communications]

Reviewers' comments:

Reviewer #1 (Remarks to the Author):

The authors report on the material Bi₁Te₁, which they studied experimentally using ARPES in conjunction with structural characterization techniques and theoretically with density functional theory. The authors claim to have identified Bi₁Te₁ as a weak topological insulator as well as a topological crystalline insulator. The analysis of the theoretical and experimental data is difficult as the surface exhibits multiple terminations with slightly different surface band structures as well as the bulk band structure has a very small band gap. Along these lines a number of questions arise, that should be addressed before publication can be recommended:

1. Relaxing a crystal structure based on van-der-Waals bonding is usually quite difficult. Although the authors say in the methods section that the relaxed structure agrees well with the experimental values, there is no real analysis given for the experimental structure determination. I would like to see a more thorough analysis (also in the main text) of the experimentally determined structure data with error bars. This is very important as the authors claim that there is a significant charge transfer between the Bi bilayers and the BiTe quintuple layers, so that small changes in the distances between the layers may have significant effects on the electronic structure, in particular the existence of a band gap.

2. The authors should comment on the communication of the surface states of the opposing surfaces. The authors used slabs between 24 and 34 layers. For pure Bi layers, this is known to be insufficient for the surfaces to be decoupled enough and quantum well states to disappear. Can the authors confirm this and discuss the relevance for the case of Bi₁Te₁? This is particularly important since the individual Bi bilayers and BiTe quintuple layers are said to be 2D TIs.

3. I am somewhat confused by the terminology used by the authors. They talk about topological insulators on one side, but then they talk about an even number of Fermi level crossings (page 6), later on about a Rashba-splitting (page 8), and on page 14 about topologically trivial states. The authors should clarify this terminology in the context of weak topological insulators.

4. Concerning the spin-polarized measurements, I also do not understand how the absence of a spin-polarization is an indication of topological character of the beta state.

5. If the authors say that they see a predominantly quintuple layer terminated surface in the ARPES data, would it not be possible to evaporate the amount of a Bi bilayer on the Bi₁Te₁ surface to distinguish the different terminations?

In summary, I am confused by the presentation, which does not really make it clear, how the authors reach their conclusions. At this stage, I cannot recommend publication in Nature Communications

Reviewer #2 (Remarks to the Author):

Typical topological insulators belong to the Z₂ class or to the topological crystalline class. The Z₂ class requires time-reversal symmetry, while the topological crystalline class requires a lattice symmetry, often a mirror symmetry. Recently, it has been reported that the prominent chalcogenide (strong) topological insulators - Bi₂Se₃ and Bi₂Te₃ - belong simultaneously to both classes; hence they are called dual topological insulators (Ref. 9).

The paper by Eschbach reports that Bi1Te1 - a stack of two quintuple layers of Bi2Te3 and one bilayer of Bi - is a dual topological insulator (TI): it is a weak and a crystalline TI. This statement originates from the theoretical part of the paper, experiments seem to support the prediction.

The computed Z2 number of (0;001) tells that the investigated (111) surface is the *dark* one (abstract), meaning that it would host no topological surface state (TSS). Being a crystalline TI as well, Bi1Te1 hosts two TSSs (the mirror Chern number equals -2; hence two TSSs). Therefore, Bi1Te1(111) appears as a conventional (singular or non-dual) TCI and the only way to check its topological nontrivial properties is to break the mirror symmetry (as has been done for example in PRB **90** (2014) 155312). It is a pity and annoying that the authors report in a single sentence on symmetry breaking in the calculations because details of these findings would support and clearly show *the* essential feature of the system.

Main results of the theory are summarized in Figure 2 which unfortunately is not very informative with respect to the major statement of the paper. I encourage to show

- zooms into the relevant energy region around the Fermi level,
- show constant energy cuts for the Fermi level, including the spin texture, and
- label the TSSs.

The discussion of the topological properties in terms of the mirror Chern numbers is by far too short. Here are two issues that need to be covered: what do modulus and sign of the mirror Chern number (MCN) tell? Why is a MCN of -2 compatible with Bi1Te1 but not with Bi2Te3?

The XRD and TEM experiments prove that one can produce Bi1Te1 samples with good quality.

Considering ARPES, the comparison of Bi1Te1 with Bi2Te3 is quite complicated (Fig. 4). The theoretical band structures for the two materials look rather similar, the arrows in 4f point to structureless grey areas which are present in 4b as well.

- What is the difference between state beta and the TSS, if there is any?
- I do not understand why details of the band structure are discussed for K-Gamma-K but not for the mirror plane M-Gamma-M. The latter is relevant with respect to the topological properties but not the former.
- I am aware that there are experts on ARPES among the authors. But one word - or better: more - on a direct comparison of ARPES data and results of DFT calculations should be in order. In particular, the spin textures may differ widely, as has been shown for Bi2Te3 and other systems.

Another comparison of the ARPES data and the DFT band structure is presented in Figure 6 and discussed on page 15.

- If the missing band gaps at higher binding energies are explained by different surface terminations (likely present in experiment), I suggest to superimpose these also in the theory panels (only data for the 1QL termination are shown).
- The state *protected by mirror symmetry* that appears at $k_x = 0$ and $k_y = 0.184$ is on a Gamma-K line which is not in a mirror plane and, therefore, not protected. Please correct (me) or discuss.

Let me summarize. The major statement of the paper is that Bi1Te1 is a dual TI, namely a weak TI and a TCI, which comes out of the DFT calculations. I have the strong impression that the experiments serve as a support for sample growth and overall electronic structure but do not help in understanding the topological nature of the system. To prove the protection one has to break the protecting symmetry which has only been done in theory (but not in experiment); it is not discussed in the paper. The paper appears thus as a follow-up in the spirit of Ref. 9 in which a dual Z2-TCI

character has been proven (for the first time?!). I do neither see need for urgent publication nor do I see an important advance of significance for the field. Therefore, I suggest to improve the paper along the issues raised and submit it to a specialized journal as a conventional article.

Some minute issues

- Title. Bi1Te1 is not well defined at this point and should be avoided. Please improve.
- Page 1.
- What is the meaning of *perfect metallic*?
- *... while other surface remain insulating.* This statement is too general because these surfaces may host trivial but metallic surface states.
- Page 4. *... $x = 0$, $y = 3$.* Should be $y = 1$?
- Figure 4. There is a lot of white space in the figure. Please try to enlarge the panels.
- Please avoid exaggerations like *very high* (page 11), *very good* (page 15).
- Page 13. Is the asymmetry function the Sherman function?

Reviewer #1 (Remarks to the Author):

The authors have addressed the criticism raised by the referees and the manuscript has become a lot more readable. I recommend publication in Nature Communications provided the following points are addressed:

1. On page 2, the last sentence before Results & Discussions: The statement that $\gamma=3$ maybe correct, but not immediately obvious. Half a sentence explaining this choice would be helpful.
2. The insets in the right panel of Fig. 2 are placed such that the states of the underlying plot make them look connected. This is confusing and should be corrected.
3. On page 4, left column, bottom paragraph, it says, "Bi-poor surfaces, ... Bi-rich surfaces": Clarify this terminology, please. Does this mean that the stoichiometry is different or is this just a matter of Te-terminated or Bi-terminated surfaces?
4. same paragraph as in 3., it says, "immediately vacuum-transferred after growth": Does this imply that the sample was transferred through air? If so, discuss how this affects the surface concerning band bending etc.
5. On page 4, right column, first paragraph, it says, "our ARPES technique lacks lateral resolution, a detailed distinction of the surface electronic features is challenging.": The authors should comment in the text on this statement more specifically, how they are able to reach their conclusions under these restrictions.

Reviewer #2 (Remarks to the Author):

I appreciate very much the detailed and enlightening response of the authors. The manuscript has considerably improved, the main statements are now well presented and explained. Thus, I recommend publication in Nature Communications.

Nevertheless, I would kindly ask the authors to improve a few minor points.

- Abstract: „... at non time-reversal invariant momenta.“ This is not well formulated: the Dirac point of a Z2-TSS is located at a TRIM, the bands themselves are not (because they are dispersive). Please consider to rephrase.

- Page 3, right column, end of the first paragraph. In my opinion, a sentence why a mirror Chern number of +2 (or -2) is incompatible with a strong TI would be helpful (odd number of band crossings of a STI).

- Page 4, right column. The theoretical bands have been shifted to higher energies. I expect that this shift is necessary because of electron correlation effects that are not described by the GGA. Please consider to add a (half-)sentence on this point.

Point-by-point response to referees:

First of all, we would like to thank the reviewers for their very detailed, thorough and constructive Reviews!

We have found valuable comments and remarks in the Reviews, which we have incorporated into the significantly improved and revised version of the manuscript. We believe that our paper presents a unique combination of theoretical prediction, epitaxial synthesis, and spectroscopic characterization of the new dual topological state (WTI + TCI), and it is worth to be presented to the broad scientific audience in Nature Communications. We believe it is important, how introducing Bi bilayers into well-known strong-TI Bi₂Te₃ leads to new topologically protected TCI states away from the TRIM points, which were clearly demonstrated in the presented spectra, and to the removal of the topological protection at the TRIM point, which is also corroborated by our spin-polarized spectra.

In the following point-by-point response to the referees we will answer and comment on every point that was raised by the referees and explain if and how the manuscript text was revised.

Reviewer #1 (Remarks to the Author):

The authors report on the material Bi₁Te₁, which they studied experimentally using ARPES in conjunction with structural characterization techniques and theoretically with density functional theory. The authors claim to have identified Bi₁Te₁ as a weak topological insulator as well as a topological crystalline insulator. The analysis of the theoretical and experimental data is difficult as the surface exhibits multiple terminations with slightly different surface band structures as well as the bulk band structure has a very small band gap. Along these lines a number of questions arise, that should be addressed before publication can be recommended:

1. Relaxing a crystal structure based on van-der-Waals bonding is usually quite difficult. Although the authors say in the methods section that the relaxed structure agrees well with the experimental values, there is no real analysis given for the experimental structure determination. I would like to see a more thorough analysis (also in the main text) of the experimentally determined structure data with error bars. This is very important as the authors claim that there is a significant charge transfer between the Bi bilayers and the BiTe quintuple layers, so that small changes in the distances between the layers may have significant effects on the electronic structure, in particular the existence of a band gap.

We agree that too little focus was put on the clear explanation of the structural relaxation in the main text and that it may be confusing to the reader which lattice constants have been used in the end. In the new version we tried to connect theoretical and experimentally deduced structural parameters better already in the main text. Our calculations using standard GGA (without taking vdW interaction into account) gave a c-lattice constant slightly larger than the experimental value (25 Å instead of 24 Å). We checked that an artificial reduction of the c-parameter to 24 Å does not change the topology and has almost no influence on the band-structure. We did now additional relaxations taking vdW interaction into account and found that the c-parameter was indeed reduced to 24 Å. We added this information in the “methods” section.

In the case of experimentally deduced lattice constants we agree that a more thorough analysis may be helpful, and therefore we moved the full characterization by X-ray diffraction and the resulting

reciprocal space maps, from which one can deduce both the in-plane and out-of-plane lattice constant a and c , from the supplementary information to Figure 3 in the main text. From this we can derive a and c accurately with an error bar. Additionally, the line profile from the STEM image (now Fig. 3c) has been improved such that actual Gaussian fits, corresponding to the individual atomic columns, are shown. From this also a lattice constant c of 23.99 +/- 0.02 Å could be deduced. We believe that this improved analysis which results in a good agreement between XRD, STEM, and the novel DFT data including van-der-Waals interactions, proves the high quality of the bulk crystal structure of our Bi1Te1 samples.

2. The authors should comment on the communication of the surface states of the opposing surfaces. The authors used slabs between 24 and 34 layers. For pure Bi layers, this is known to be insufficient for the surfaces to be decoupled enough and quantum well states to disappear. Can the authors confirm this and discuss the relevance for the case of Bi1Te1? This is particularly important since the individual Bi bilayers and BiTe quintuple layers are said to be 2D TIs.

To exclude the possibility that the band gaps along G-K are caused by cross-talk of the film surfaces, we calculated 72 and 82 layer films of QL and 2QL terminated Bi1Te1 using Wannier functions extracted from the DFT results. As shown now in the supplement this does not close the gaps, indicating that we indeed have a WTI (as also predicted by the analysis of the parities of the bulk band-structure).

3. I am somewhat confused by the terminology used by the authors. They talk about topological insulators on one side, but then they talk about an even number of Fermi level crossings (page 6), later on about a Rashba-splitting (page 8), and on page 14 about topologically trivial states. The authors should clarify this terminology in the context of weak topological insulators.

We agree, that the situation can be a bit confusing in the dual TCI/WTI presented here: On the dark surface of a WTI (the one investigated in our case) it is required to have an even number of Fermi level crossings, otherwise it would be a normal TI. Nevertheless, this surface is not dark (i.e. having zero crossings) in our case, but obtains two crossings from the TCI character of BiTe. Independent of the topological properties, surface states on nonmagnetic materials containing heavy atoms are more or less spin-split due to the Rashba effect. In TIs, these states are often referred to as topologically trivial, since these states are also present when the topology is changed to trivial, e.g. by tuning (in the calculation) a parameter such as spin-orbit coupling. We explain the differences now more carefully in the main text.

4. Concerning the spin-polarized measurements, I also do not understand how the absence of a spin-polarization is an indication of topological character of the beta state.

To be on the safe side, let us first clarify what Referee has in mind in his/her comment. We claim that the absence of spin polarization may be an indication of **topologically trivial** character of that state near the Fermi level in Bi1Te1, centered at the Gamma-bar point.

ARPES spectra of Bi2Te3 and Bi1Te1 look relatively similar, but their experimentally obtained spin textures near the Fermi level are very different, with vanishing polarization in case of Bi1Te1. This is especially unexpected taking into account that the polarization of deeper lying states is very similar in Bi2Te3 and Bi1Te1. Moreover, these findings indicate that in the field of topological insulators one

has to be very careful because e.g. electronic states that on first glance look like topologically non-trivial Dirac cone states can in fact be of trivial origin.

5. If the authors say that they see a predominantly quintuple layer terminated surface in the ARPES data, would it not be possible to evaporate the amount of a Bi bilayer on the Bi₁Te₁ surface to distinguish the different terminations?

In general, we agree with the referee and think that performing this experiment in ideal fashion would be helpful, however:

1) It is experimentally challenging to evaporate a perfect bilayer and definitely requires much more experimental work beyond the scope of the present manuscript,

2) The influence of a Bi BL on the surface of Bi₂Te₃ substrates and similar experiments were extensively performed by T. Hirahara et al. (Ref. 22) or L. Miao et al. (Ref. 23),

3) The deposition of the right amount of a Bi material for a bilayer onto the Bi₁Te₁ substrate which exhibits a still unknown morphology of differently terminated regions will likely not result in a homogenous Bi bilayer-termination. Instead, we expect an even more complex distribution of different terminations, such as BL-QL-BL or QL-BL-BL.

Therefore, we postpone such an experiment for future studies.

In summary, I am confused by the presentation, which does not really make it clear, how the authors reach their conclusions. At this stage, I cannot recommend publication in Nature Communications

We believe that in the revised manuscript we were able to substantially improve the clarity of the presentation. We hope that current version can be accepted to Nature Communications.

Reviewer #2 (Remarks to the Author):

Typical topological insulators belong to the Z₂ class or to the topological crystalline class. The Z₂ class requires time-reversal symmetry, while the topological crystalline class requires a lattice symmetry, often a mirror symmetry. Recently, it has been reported that the prominent chalcogenide (strong) topological insulators - Bi₂Se₃ and Bi₂Te₃ - belong simultaneously to both classes; hence they are called dual topological insulators (Ref. 9).

The paper by Eschbach reports that Bi₁Te₁ - a stack of two quintuple layers of Bi₂Te₃ and one bilayer of Bi - is a dual topological insulator (TI): it is a weak and a crystalline TI. This statement originates from the theoretical part of the paper, experiments seem to support the prediction.

The computed Z₂ number of (0;001) tells that the investigated (111) surface is the *dark* one (abstract), meaning that it would host no topological surface state (TSS). Being a crystalline TI as well, Bi₁Te₁ hosts two TSSs (the mirror Chern number equals -2; hence two TSSs). Therefore, Bi₁Te₁(111) appears as a conventional (singular or non-dual) TCI and the only way to check its topological nontrivial properties is to break the mirror symmetry (as has been done for example in PRB **90** (2014) 155312). It is a pity and annoying that the authors report in a single sentence on symmetry

breaking in the calculations because details of these findings would support and clearly show *the* essential feature of the system.

First of all, we would like to clarify our used terminology. In our manuscript we use the term “dual” topological insulator to describe a weak topological insulator plus a topological crystalline insulator (WTI + TCI). This is similar but not identical to the situation in our Ref. 8 (PRL 112, 016802 (2014)) where Rauch et al. for the first time use this term for the simultaneous character of Bi₂Te₃ as a strong TI + TCI. Nevertheless, we adapted this term in our case for Bi₁Te₁, which is a WTI + TCI.

In order to clarify this adapted but modified terminology, a sentence has been included in the introduction part of the manuscript. We also included a panel in Fig. 2 that shows the effect of mirror symmetry breaking: as expected, the band gap opens along the G-M line when the mirror symmetry is broken in that direction.

Main results of the theory are summarized in Figure 2 which unfortunately is not very informative with respect to the major statement of the paper. I encourage to show

- zooms into the relevant energy region around the Fermi level,
- show constant energy cuts for the Fermi level, including the spin texture, and
- label the TSSs.

We agree that the important features were not clearly seen in Fig. 2; we replaced the plots by zooms into the relevant regions and focus on the QL and 2QL termination. The old Fig.2, which gives an overview over a larger energy range and the bulk projected states is shifted to the supplement. There, we also discuss the constant energy cuts and the spin-texture. We labeled the state which opened a gap after mirror symmetry breaking as TCI-state in Fig. 2.

The discussion of the topological properties in terms of the mirror Chern numbers is by far too short. Here are two issues that need to be covered: what do modulus and sign of the mirror Chern number (MCN) tell? Why is a MCN of -2 compatible with Bi₁Te₁ but not with Bi₂Te₃?

We extended the discussion of the mirror Chern numbers, mentioning the relation between modulus and number of Fermi level crossing as well as the connection between its sign and the spin-orientation. We also cited the appropriate literature. As for the (in)compatibility of MCN=-2 with Bi₂Te₃, we note that in Ref. [8] the MCN of Bi₂Te₃ was determined to be -1.

The XRD and TEM experiments prove that one can produce Bi₁Te₁ samples with good quality.

Considering ARPES, the comparison of Bi₁Te₁ with Bi₂Te₃ is quite complicated (Fig. 4). The theoretical band structures for the two materials look rather similar, the arrows in 4f point to structureless grey areas which are present in 4b as well.

Following Referees's suggestion we have enhanced Fig.4 considerably.

The newly made Fig. 4 now shows the wide-energy range ARPES spectra of the two materials next to the interesting EDCs (old panel (i)) which are best suitable for the direct comparison. The spectral differences become very clear in these two EDCs taken at different k_{\parallel} values. The way of presenting this has been improved in our opinion.

Further, we changed Fig. 4 (b) and (c) completely to more careful and detailed comparison of the magnified electronic structure close to the Fermi level between experiment and theory. Panels (b) and (c) now contain a step-by-step comparison of the ARPES spectra (along both GK and GM) and the respective DFT calculations, plus a final superposition of the spectra. We believe that this way of presenting has substantially improved the readability of the figure.

- What is the difference between state beta and the TSS, if there is any?

We believe that new Fig.4 (b) and (c) show the difference between those states better and also the fact that the “beta” state (which is not any more called like this in the new manuscript version) very likely does not consist of only one state but a merged superposition (or hybridization) of several states.

In addition, Fig. 5 also shows a sizable difference in the spin polarization of these states close to the Fermi level, which demonstrates another considerable difference in our opinion.

- I do not understand why details of the band structure are discussed for K-Gamma-K but not for the mirror plane M-Gamma-M. The latter is relevant with respect to the topological properties but not the former.

We followed the suggestion of the Referee and Fig. 4 now shows both directions. However, the detailed comparison in Fig. 6 is still for Gamma-M. As visible in Fig. 2, the TCI-Dirac cone is strongly tilted along Gamma-M such that its Dirac crossing is hard to resolve. In contrast, upright Dirac cones with the crossing point on the Gamma-M line are available perpendicular to Gamma-M.

- I am aware that there are experts on ARPES among the authors. But one word - or better: more - on a direct comparison of ARPES data and results of DFT calculations should be in order. In particular, the spin textures may differ widely, as has been shown for Bi₂Te₃ and other systems.

In our interpretation spin-integrated ARPES measured initial state dispersions. Intensity of spectral features is weighted by photon-energy dependent matrix element effects, and to take it into account we present spectra at two different photon energies (now shown in the supplement).

Interpretation of spin-polarized data is more complex. We show, that under the same conditions (photon energy, light polarization, experimental geometry) spin polarized spectra for Bi₂Te₃ and Bi₁Te₁ are similar for higher binding energy valence states, but different close to the Fermi level. This allowed us to conclude that the spin-textures of the states near the Fermi level are different for the two materials.

Another comparison of the ARPES data and the DFT band structure is presented in Figure 6 and discussed on page 15.

- If the missing band gaps at higher binding energies are explained by different surface terminations (likely present in experiment), I suggest to superimpose these also in the theory panels (only data for the 1QL termination are shown).

In the supplement we have added experimental dispersions at He I 21.22 eV excitation, where spectral weight in the interesting region between 400 and 600 meV is strongly suppressed. In the spectra at Xe 8.4 eV excitation the experimental spectral weight in the interesting regions between 400 and 600 meV is also decreased, however, one can still identify dispersive features crossing this region in experimental spectra.

Following Referee's comment, we have performed additional calculations and added a short paragraph to address this issue (last paragraph before "Summary and conclusions") and in the supplement we have added theoretical dispersions for the 2QL and BL terminations. Local LDA gaps are still present for the case of 2QL termination, but they are not present for the case of Bi bilayer termination.

Summing up these observations we only slightly modify our initial interpretation. It is unavoidable to have different terminations, which lead to the appearance of new dispersive features, and the cross section of these features is strong at 8.4 eV photon energy, where they can be observed in the E(k) maps, while at 21.22 eV they are hardly visible.

- The state *protected by mirror symmetry* that appears at $k_x = 0$ and $k_y = 0.184$ is on a Gamma-K line which is not in a mirror plane and, therefore, not protected. Please correct (me) or discuss.

Here the referee is mistaken. In Fig. 6 we do indeed show ARPES spectra along directions that are parallel to Gamma-K but the interesting crossing point of course lies on the Gamma-M line.

The crossing "point" can be seen always along two directions. For clarification Fig.6 was also modified and in (a) a sketch of the surface BZ was added.

Let me summarize. The major statement of the paper is that Bi1Te1 is a dual TI, namely a weak TI and a TCI, which comes out of the DFT calculations. I have the strong impression that the experiments serve as a support for sample growth and overall electronic structure but do not help in understanding the topological nature of the system. To prove the protection one has to break the protecting symmetry which has only been done in theory (but not in experiment); it is not discussed in the paper. The paper appears thus as a follow-up in the spirit of Ref. 9 in which a dual Z2-TCI character has been proven (for the first time?!). I do neither see need for urgent publication nor do I see an important advance of significance for the field. Therefore, I suggest to improve the paper along the issues raised and submit it to a specialized journal as a conventional article.

In recent years, extensive theoretical efforts have led to a number of predictions of non-trivial topological phases. The key advantage of our study is in presenting experimental evidence of the predicted complex topology. In particular, TCI nodes away from TRIM points are experimentally identified, while the properties of the near-Fermi level TRIM-centered spin textures show that this state which bears a resemblance to the TSS of Bi2Te3, has been rendered trivial in Bi1Te1. We believe that these findings add substantially to the field of complex topological materials, and show how minute changes in the crystal structure can dramatically influence topological band properties. Thereby, we open the field to play with the different protecting symmetries in order to switch conductivity on protected planes.

Some minute issues

- Title. Bi1Te1 is not well defined at this point and should be avoided. Please improve.

We like the title as it is and hope that the referee will understand this as our creative right. We agree and are aware of the fact that the term Bi1Te1 is not yet precisely defined at this point but since we are explaining in the first sentence of the abstract exactly what it means and what material the paper is all about, we hope the referee will accept this.

- Page 1.

- What is the meaning of *perfect metallic*?

“perfect” was removed.

- *... while other surface remain insulating.* This statement is too general because these surfaces may host trivial but metallic surface states.

We agree with the referee and describe the influence of topologically protected surface states only, in the new version of the manuscript.

- Page 4. *... $x = 0$, $y = 3$.* Should be $y = 1$?

No. Here the terminology of Bi₂Te₃ is indeed $x=0$ and $y=3$ because a unit cell of Bi₂Te₃ consists of 3 QLs (and no BL).

- Figure 4. There is a lot of white space in the figure. Please try to enlarge the panels.

Figure 4 was changed completely.

- Please avoid exaggerations like *very high* (page 11), *very good* (page 15).

We tried to avoid those words in the new version.

- Page 13. Is the asymmetry function the Sherman function?

This is correct. We added this information, although strictly speaking the Sherman function was defined for Mott detectors only. Since we are using a less common SPLEED detector, we believe the term asymmetry function is more generally correct.

We would like to thank the Reviewers for their positive comments. We have edited manuscript according to the suggestions of the Reviewers. Below we provide detailed comments on the changes.

Reviewer #1 (Remarks to the Author):

The authors have addressed the criticism raised by the referees and the manuscript has become a lot more readable. I recommend publication in Nature Communications provided the following points are addressed:

1. On page 2, the last sentence before Results & Discussions: The statement that $y=3$ maybe correct, but not immediately obvious. Half a sentence explaining this choice would be helpful.

We have added this explanation.

2. The insets in the right panel of Fig. 2 are placed such that the states of the underlying plot make them look connected. This is confusing and should be corrected.

We have prepared a new version of Fig. 2 where this has been corrected.

3. On page 4, left column, bottom paragraph, it says, "Bi-poor surfaces, ... Bi-rich surfaces": Clarify this terminology, please. Does this mean that the stoichiometry is different or is this just a matter of Te-terminated or Bi-terminated surfaces?

It is a matter of Te- and Bi-terminated surfaces. We added additional explanation.

4. same paragraph as in 3., it says, "immediately vacuum-transferred after growth": Does this imply that the sample was transferred through air? If so, discuss how this affects the surface concerning band bending etc.

Samples measured in the lab-based ARPES were never exposed to air, they were always kept under UHV conditions, we have used vacuum suitcase. This concerns all the $E(k)$ spectra presented in the paper. Samples measured by spin-ARPES in DELTA/Dortmund were exposed to air and they were cleaned by sputtering and annealing. This concerns spin-polarized EDCs shown in Fig. 5 (b)-(c) and (e)-(f).

5. On page 4, right column, first paragraph, it says, "our ARPES technique lacks lateral resolution, a detailed distinction of the surface electronic features is challenging.": The authors should comment in the text on this statement more specifically, how they are able to reach their conclusions under these restrictions.

We believe that this is already well explained in the paper. More than one type of termination is always present on the surface, therefore due to beam spot size of 400 micrometers we always probe a superposition on different terminations, even though one of the terminations dominate. We replaced “lacks lateral resolution” by “averages over the beam spot” size to make our statement more clear.

Reviewer #2 (Remarks to the Author):

I appreciate very much the detailed and enlightening response of the authors. The manuscript has considerably improved, the main statements are now well presented and explained. Thus, I recommend publication in Nature Communications.

Nevertheless, I would kindly ask the authors to improve a few minor points.

- Abstract: „... at non time-reversal invariant momenta.“ This is not well formulated: the Dirac point of a Z2-TSS is located at a TRIM, the bands themselves are not (because they are dispersive). Please consider to rephrase.

We have changed to abstract according to the rules of Nature Comm., as requested by the Editor. This formulation has been rephrased.

- Page 3, right column, end of the first paragraph. In my opinion, a sentence why a mirror Chern number of +2 (or -2) is incompatible with a strong TI would be helpful (odd number of band crossings of a STI).

We have added the sentence in this paragraph as requested.

- Page 4, right column. The theoretical bands have been shifted to higher energies. I expect that this shift is necessary because of electron correlation effects that are not described by the GGA. Please consider to add a (half-)sentence on this point.

We don't think correlations play a major role on this energy scale. Correlations are known to only slightly renormalize the dispersion of Dirac cones. These shifts are necessary due to the intrinsic doping of as-grown epilayers by vacancies and antisite defects, which in both cases produces n-type material, i.e. the Fermi level is cutting through the conduction band. We left the original version.